# Prospective multicentre head-to-head validation of host blood transcriptomic biomarkers for pulmonary tuberculosis by real-time PCR

Simon C. Mendelsohn [1], Stanley Kimbung Mbandi [1], Andrew Fiore-Gartland[2], Adam Penn-Nicholson [1], Munyaradzi Musvosvi[1], Humphrey Mulenga[1], Michelle Fisher[1], Katie Hadley[1], Mzwandile Erasmus[1], Onke Nombida[1], Michèle Tameris[1], Gerhard Walzl [3], Kogieleum Naidoo[4,5], Gavin Churchyard[6,7,8], Mark Hatherill[1] & Thomas J. Scriba [1✉]

## Abstract

**Background** Sensitive point-of-care screening tests are urgently needed to identify individuals at highest risk of tuberculosis. We prospectively tested performance of host-blood transcriptomic tuberculosis signatures.

**Methods** Adults without suspicion of tuberculosis were recruited from five endemic South African communities. Eight parsimonious host-blood transcriptomic tuberculosis signatures were measured by microfluidic RT-qPCR at enrolment. Upper respiratory swab specimens were tested with a multiplex bacterial-viral RT-qPCR panel in a subset of participants. Diagnostic and prognostic performance for microbiologically confirmed prevalent and incident pulmonary tuberculosis was tested in all participants at baseline and during active surveillance through 15 months follow-up, respectively.

**Results** Among 20,207 HIV-uninfected and 963 HIV-infected adults screened; 2923 and 861 were enroled. There were 61 HIV-uninfected (weighted prevalence 1.1%) and 10 HIV-infected (prevalence 1.2%) tuberculosis cases at baseline. Parsimonious signature diagnostic performance was superior among symptomatic (AUCs 0.85–0.98) as compared to asymptomatic (AUCs 0.61–0.78) HIV-uninfected participants. Thereafter, 24 HIV-uninfected and 9 HIV-infected participants progressed to incident tuberculosis (1.1 and 1.0 per 100 person-years, respectively). Among HIV-uninfected individuals, prognostic performance for incident tuberculosis occurring within 6–12 months was higher relative to 15 months. 1000 HIV-uninfected participants were tested for respiratory microorganisms and 413 HIV-infected for HIV plasma viral load; 7/8 signature scores were higher ($p < 0.05$) in participants with viral respiratory infections or detectable HIV viraemia than those without.

**Conclusions** Several parsimonious tuberculosis transcriptomic signatures met triage test targets among symptomatic participants, and incipient test targets within 6 months. However, the signatures were upregulated with viral infection and offered poor specificity for diagnosing sub-clinical tuberculosis.

## Plain Language Summary

Delays in tuberculosis (TB) diagnosis result in increased disease severity, risk of death, and infection of further individuals. The presence of symptoms is typically used to find people with TB. However, about half of individuals with TB are asymptomatic. We evaluated blood samples from individuals living in areas with high incidence of TB to see whether there were changes in components of the blood following infection with TB. Markers were identified that diagnosed TB in symptomatic adults, but were not as accurate to detect TB in those without symptoms. Most markers tested were able to accurately predict progression to TB within 6 months in HIV-uninfected individuals. These markers in the blood could enable the screening of symptomatic adults and predict TB risk, thus enabling targeting of therapy.

A full list of author affiliations appears at the end of the paper.

An estimated 2.9 million people with tuberculosis (TB) disease went undiagnosed or unreported in 2019[1], partly because passive case-finding strategies rely primarily on self-presentation of people for healthcare when they experience TB symptoms, and dependence on sputum for microbiological testing. TB prevalence surveys in Africa and Asia demonstrate that approximately 50% of TB cases are asymptomatic (subclinical), highlighting a major blind spot in the symptom-dependent case-finding strategies, which may perpetuate *Mycobacterium tuberculosis* (Mtb) transmission[2–5]. Currently available triage tests and passive case-finding strategies for symptomatic (clinical) disease are thus inadequate to find these missing cases among people who do seek healthcare. Effective active case-finding tools for screening of asymptomatic individuals in the community are urgently needed for earlier detection, further microbiological testing, and treatment, in order to interrupt the spread of Mtb.

Computer-aided detection for automated interpretation of digital chest radiographs[6] is one such approach for rapid and cheap mass screening, particularly in resource-constrained settings. Mobile sputum Xpert MTB/RIF screening also performs well as a community-based case-finding strategy, but requires considerable resources and is dependent on availability of a sputum sample[7]. Tests for detection of Mtb-exposed individuals at high risk of progression to TB disease, particularly in high burden settings, are also urgently needed to guide TB preventive therapy (TPT). Interferon-γ release assays (IGRAs) and tuberculin skin testing lack specificity in endemic settings due to high rates of prior Mtb exposure, which result in both overtreatment of those unlikely to progress to disease and missed incipient cases[8]. IGRAs are also laborious, and require laboratory facilities and skilled technicians.

We previously developed a predictive correlate of risk signature, RISK11, for diagnosis of TB disease and for identification of individuals at high risk of progression to active TB disease[9,10]. Diagnostic and prognostic performance of RISK11 as a PCR-based screening test was prospectively validated in HIV-uninfected adults[11] and people living with HIV (PLHIV)[12]. This signature is not ideal for further development due to its large size (48 primer-probe assays). Fortunately, several refined, parsimonious diagnostic and prognostic transcriptomic signatures, which are more amenable to translation to point-of-care testing, have been developed[13–16]. However, most transcriptomic signature discovery and validation studies use a case-control design with stringent exclusion criteria, resulting in spectrum bias and best-case discriminative performance, which is not reproducible in prospective studies in heterogenous real-world settings. Transcriptomic signatures are predominantly measured by microarray or RNA-sequencing, techniques which are not suitable for use at the point of care. Current PCR methods are also laborious and require a laboratory and skilled operators. However, the successful development of the fingerstick blood-based Cepheid Xpert-MTB-Host Response (HR)-Prototype cartridge system[17–19] clearly demonstrates proof-of-concept that a near-point-of-care platform can be implemented. There are also few studies which have performed unbiased head-to-head comparisons of signature performance[13–15] and none for active case-finding in a community setting. It is therefore not clear which, if any, signatures offer potential in this context and should be advanced along the developmental pipeline into a near-point-of-care platform.

In this paper we report the results of a multicentre observational study, which prospectively tested the performance of eight parsimonious host-blood TB interferon-stimulated gene (ISG) signatures measured by microfluidic real-time quantitative PCR (RT-qPCR), for diagnosing prevalent pulmonary TB and predicting progression to incident TB disease, in a predominantly asymptomatic community cohort of HIV-uninfected and HIV-infected South African adults. As secondary aims, we compared signature diagnostic performance in symptomatic and asymptomatic participants, and benchmarked parsimonious signature accuracy against RISK11 and the World Health Organization (WHO) Target Product Profile (TPP) criteria for triage and prognostic TB tests[20,21]. As an exploratory aim, we also explored the effect of HIV and upper respiratory tract microorganisms on transcriptomic signature scores.

We found that diagnostic performance of the parsimonious signatures was superior among symptomatic as compared to asymptomatic participants; While none of the signatures met WHO triage test TPP criteria to diagnose subclinical TB, several signatures met or approached the criteria for diagnosing clinical TB. Most signatures had a good prognostic performance for incident TB occurring within 6 months of testing. Signature scores for most signatures were higher in the HIV-infected participants with detectable HIV plasma viral load, or HIV-uninfected participants with upper respiratory tract viruses, but not bacteria, as compared to those with undetectable HIV viral load or no respiratory viruses, respectively. Accordingly, most signatures were unable to differentiate individuals with TB from those with those with viral upper respiratory tract infection. Parsimonious TB transcriptomic signatures hold promise for triage of symptomatic adults seeking care, screening of ART clinic attendees, and prediction of short-term risk of TB for initiation of targeted preventive therapy. However, ISG induction from common viral infections results in poor specificity which is problematic for community active case-finding efforts in endemic settings, where there is a high burden of undiagnosed subclinical disease.

## Methods

**Study design and participants**. In this diagnostic and prognostic accuracy study, we evaluated parsimonious blood-based host transcriptomic signature performance in four South African cohorts (Supplementary Fig. S1). We first translated the signatures to a RT-qPCR platform and validated performance in a case-control study of HIV-uninfected and HIV-infected individuals with and without pulmonary TB, the Cross-sectional TB Cohort (CTBC) study (Supplementary Fig. S1a). We then prospectively tested signature performance in HIV-uninfected volunteers enroled in the Correlate of Risk Targeted Intervention Study (CORTIS-01; ClinicalTrials.gov: NCT02735590; Supplementary Fig. S1b) and PLHIV enroled in the observational CORTIS-HR study (Supplementary Fig. S1d). Finally, we evaluated the effects of upper respiratory microorganisms on transcriptomic signatures in a subset of HIV-uninfected participants screened for the CORTIS-01 trial (Supplementary Fig. S1c). This study is reported in accordance with the Standards for Reporting Diagnostic Accuracy Studies initiative guidelines[22].

**Ethical approval**. The study protocol was approved by the University of Cape Town Faculty of Health Sciences Human Research Ethics Committee (HREC 812/2017) and institutional human research ethics committees at each participating site. Written informed consent was obtained from all participants and all experiments conformed to the principles set out in the WMA Declaration of Helsinki and the US Department of Health and Human Services Belmont Report.

**Cross-sectional TB Cohort (CTBC)**. The CTBC case-control study was previously described[23]. Briefly, adults aged ≥18 years with newly diagnosed pulmonary TB confirmed by sputum Xpert

MTB/RIF (Cepheid, Sunnyvale, CA, USA) and/or liquid myco-bacterial culture (Mycobacteria Growth Indicator Tube [MGIT], BACTEC, Beckton Dickinson, Franklin Lakes, NJ, USA) were enroled at primary healthcare clinics in Worcester and Masi-phumelele, South Africa. Healthy, asymptomatic community controls were recruited from the same areas. HIV infection was diagnosed with the Determine HIV1/2 test (Alere, Waltham, MA, USA). All participants provided written, informed consent and the protocols were approved by the University of Cape Town Faculty of Health Sciences Human Research Ethics Committee (HREC 126/2006 and 288/2008).

**Correlate of risk-targeted intervention study (CORTIS).** The CORTIS-01 clinical trial in HIV-uninfected individuals and CORTIS-HR observational study in PLHIV have previously been described[11,12]. Briefly, healthy adult volunteers without clinical suspicion of TB, residing in five TB endemic communities in South Africa (Durban, Klerksdorp, Ravensmead, Rustenburg, and Worcester), were recruited through word-of-mouth, house-to-house visits, and liaison with non-governmental organisations. Recruitment did not target symptomatic individuals seeking healthcare or other high-risk groups. Eligible participants aged 18–59 years were without comorbidities (except for HIV) and did not have known TB disease, or household exposure to individuals with multi-drug resistant TB, within the prior three years. The RISK11 signature was measured at enrolment, and a pre-specified score threshold of 60% was used to differentiate RISK11-positive from RISK11-negative participants. The HIV-uninfected COR-TIS-01 population was enriched for participants at high risk of TB disease through enrolment of all eligible RISK11-positive individuals, and randomisation of RISK11-negative participants to enrolment or non-enrolment at a 1:7–9 ratio. RISK11-positive participants were randomised to receive TPT (weekly high-dose isoniazid and rifapentine for 3 months, 3HP) or no intervention; RISK11-negative participants were randomised to no intervention or excluded from the study. In the CORTIS-HR observational study, PLHIV were enroled irrespective of RISK11 status and referred for standard of care antiretroviral and isoniazid pre-ventive therapy.

All CORTIS participants provided two spontaneously expecto-rated sputum samples, if able to, at enrolment and at the end of study visits. End of study visits were performed at month 15 of follow-up or at an earlier timepoint for withdrawn participants. Both samples collected at enrolment from HIV-uninfected participants (CORTIS-01) were tested for Mtb using Xpert MTB/RIF, and from HIV-infected participants (CORTIS-HR) using Xpert MTB/RIF and MGIT culture. The two samples collected at the end of study visit were tested using Xpert MTB/RIF or Xpert Ultra (Cepheid), and MGIT culture. In addition, symptom-triggered TB investigations (two sputum samples; one for Xpert MTB/RIF and one for MGIT culture) were performed at six routine study visits through 15 months follow-up. TB symptoms included at least one of: persistent unexplained cough, night sweats, fever, or weight loss for 2 weeks or more, or any haemoptysis. Participants diagnosed with microbiologically-confirmed TB were withdrawn from the study and referred for curative treatment.

**Respiratory pathobionts cohort.** The Respiratory Pathobionts cohort, a subset of participants screened for eligibility for the CORTIS-01 trial at the Worcester site, has previously been described[24]. Briefly, HIV-uninfected participants were con-secutively enroled in this sub-study irrespective of CORTIS-01 enrolment, or signs and symptoms of upper respiratory tract infections. Paired nasopharyngeal and oropharyngeal flocked swabs (FLOQSwabs, Copan Diagnostics, Murrieta, CA, USA) were collected and stored in Primestore buffer (Longhorn Vac-cines and Diagnostics, San Antonio, TX, USA) at −80 °C, and viral and bacterial nucleic acid was later extracted (Qiasymphony Virus/Bacteria Mini Kit, Qiagen, Hilden, Germany) and quanti-fied using a multiplex RT-qPCR assay kit (Respiratory Pathogens 33 Kit, Fast Track Diagnostics, Luxembourg) on the CFX96 Touch System lightcycler platform (Bio-Rad, Hercules, CA, USA), according to the manufacturer's instructions. Participants co-enroled into the CORTIS-01 trial were investigated for TB at baseline; those only enroled into the Respiratory Pathobionts cohort were not investigated for TB (Supplementary Fig. S1c). All participants provided written, informed consent and the proto-cols were approved by the University of Cape Town Faculty of Health Sciences Human Research Ethics Committee (HREC 327/2017).

**Blood sample collection and RNA extraction.** Venous whole blood was collected in PAXgene tubes (PreAnalytiX, Hom-brechtikon, Switzerland) at enrolment in all cohorts, frozen at −20 °C, and shipped to the South African Tuberculosis Vaccine Initiative (SATVI) Cape Town laboratory. For the CTBC study, RNA was manually extracted with the PAXgene blood RNA kit (Qiagen) according to the manufacturer's instructions, stored at −80 °C, and later used for transcriptomic analysis. For the other cohorts, RNA was extracted using a high-throughput, standar-dised, and reproducible fully automated procedure on the Free-dom EVO 150 robotic platform (Tecan, Männedorf, Switzerland) with the Maxwell SimplyRNA kit (Promega, Madison, WI, USA). One aliquot of RNA was used immediately for cDNA synthesis and measurement of the RISK11 signature, and a second aliquot was stored at −80 °C, and later used to measure the panel of parsimonious transcriptomic signatures.

**Measurement of TB transcriptomic signatures.** We translated measurement of eight parsimonious TB transcriptomic signatures —Francisco2[25], Maertzdorf4[26] (also known as DIAG4), Penn-Nicholson6 (also known as RISK6)[23], Suliman4 (also known as RISK4)[27], Roe1[28] (BATF2 only), Roe3[29], Sweeney3[30], and Thompson5[31] (also known as RESPONSE5)—to a microfluidic RT-qPCR platform using TaqMan primer-probe assays for quantification of transcripts, and signature scores were calculated as previously reported (Tables S1–6 in Supplementary Data 1). The Herberg2 signature[32], designed to discriminate between viral and bacterial infection in febrile children, was also included as a potential discriminator between viral and mycobacterial infection in the Respiratory Pathobionts cohort. The signatures were pragmatically selected in 2017 for inclusion in this head-to-head validation alongside RISK11 based on the availability of validated performance data, number of transcripts (six or fewer), and availability of target sequences for custom primer-probe design or predesigned TaqMan assay.

Following cDNA synthesis with EpiScript reverse transcriptase (Lucigen, Middleton, WI, USA), genes of interest (Tables S2–S3 in Supplementary Data 1) were pre-amplified using pools of TaqMan primer-probe assays (Thermo Fisher Scientific, Wal-tham, MA, USA), and gene expression (raw cycle threshold, Ct) quantified by microfluidic multiplex RT-qPCR using either Fluidigm (San Francisco, CA, USA) 96.96 (96 samples multi-plexed with 96 primer-probe assays) or 192.24 (192 samples multiplexed with 24 primer-probe assays) Gene Expression chips on the BioMark HD instrument (Fluidigm).

**RT-qPCR primer-probe assay panel design.** The RISK11 tran-scriptomic signature was measured as previously described with

pre-qualified TaqMan gene expression primer-probe assays[9–11]. The Roe1 and Roe3 signatures scores were calculated from the same Fluidigm 96.96 gene expression chip assay panels as the RISK11 signature (Table S2 in Supplementary Data 1) by subtracting the geometric mean of four housekeeping gene raw Ct values from the average raw Ct of the genes of interest (Table S1 in Supplementary Data 1).

A panel of 24 TaqMan gene expression primer-probe assays for the other seven signatures (Table S3 in Supplementary Data 1)—Francisco2, Herberg2, Maertzdorf4, Penn-Nicholson6 (Table S4 in Supplementary Data 1), Suliman4 (Table S5 in Supplementary Data 1), Sweeney3, and Thompson5 (Table S6 in Supplementary Data 1)—were run in parallel on Fluidigm 192.24 gene expression chips. Primer-probe assays were qualified to ensure efficient amplification, linearity, and multiplexing capability, using methods similar to those described by Dominguez et al.[33] Briefly, we used five pre-amplified cDNA samples of varying concentration with a 12-point, two-fold dilution series. Each of the five dilution series were divided into eight segments of five consecutive dilutions and analysed for linearity in an iterative manner using the linear-least squares regression between the logarithm of the cDNA concentration and amplification cycle (Et; 40 – Ct). An assay passed qualification if at least one segment within a dilution series met the following three criteria: (1) strong correlation (Pearson $r^2 \geq 0.99$), (2) linear slope (regression coefficient between 3.1–3.6), and (3) efficient amplification ($[10^{1/slope} – 1] = 90\text{–}110\%$). All new primer-probe assays passed qualification with efficient linear amplification (Table S3 in Supplementary Data 1).

**Data quality control and analysis**. Fluidigm 192.24 gene expression chips were analysed using a locked-down R script (bitbucket.org/satvi/sixs) with quality control filters that assessed the integrity and reproducibility of each chip. The following parameters were applied for extracting Ct values: Linear (Derivative) baseline correction, Quality Threshold of 0.3, and Auto (Global) for Ct Threshold Method using Fluidigm Biomark software version 4.5.1. Chips with marked deviation in internal positive control sample primer-probe assay Ct values or RISK6 signature score from historical runs, with detection in the no-template (water) control or no-reverse-transcriptase (to detect amplification of genomic DNA) control in primer-probes spanning exon-exon junctions, or with more than 20% failed reactions for a particular primer-probe assay, were repeated. Individual samples with more than 20% failed primer-probe reactions were classified as failed and no signature scores were computed. If less than 20% of primer-probe reactions failed for an individual sample, signature scores were computed where possible and signatures with missing primer-probe raw Ct values were deemed failed for that sample. Samples and primer-probe assays were run in singlet, and failed signature results for individual samples were assumed to follow a random distribution, thus not repeated, and excluded from analysis.

**Endpoints**. For the CTBC study and Respiratory Pathobionts sub-study, TB disease was defined by a single sputum sample positive for Mtb on either Xpert MTB/RIF and/or liquid culture at enrolment. The coprimary protocol-specified endpoints in the CORTIS studies were baseline prevalent TB disease and incident disease through 15 months follow-up confirmed by a positive Xpert MTB/RIF, Xpert Ultra, or MGIT culture, on two or more separate sputum samples collected within any 30-day period. The secondary endpoint was microbiologically confirmed TB disease on at least one sputum sample. All Xpert Ultra trace positive results were excluded from the analysis because of the risk of false positives.

**Statistics and reproducibility**. All primary and secondary analyses for the CORTIS-01 and CORTIS-HR sub-studies were pre-specified in the statistical analysis plan, but not in the original parent study protocols. CTBC and Respiratory Pathobionts sub-study analyses were exploratory. All statistical analyses were done in R (Boston, MA, USA), version 3.6.1. Sample sizes for this sub-study were not pre-specified, but were based on the accumulated enrolment of the CTBC and CORTIS parent studies. Performance of the signatures was evaluated on their ability to diagnose prevalent TB disease at baseline and predict progression to incident TB disease through 15-month follow-up. For all analyses, HIV-infected (CORTIS-HR) and HIV-uninfected (CORTIS-01) participants were considered independently. Diagnostic performance at baseline was assessed in all participants enroled in the CORTIS studies with available PAXgene samples. Thereafter, participants meeting the primary prevalent TB endpoint definition, participants who did not attend further follow-up visits, and participants randomised to the RISK11-positive 3HP group (CORTIS-01 only) were excluded from the primary endpoint prognostic performance analysis. Participants were followed for up to 15 months; those who discontinued follow-up prior to 15 months and did not meet the primary endpoint definition were censored at their final study visit or last negative sputum sample collection but included in the prognostic analysis.

Due to enrichment of RISK11-positive individuals in the CORTIS-01 enroled population, analyses of signature performance required participant weighting to obtain estimates applicable to the screened population, effectively upweighting RISK11-negative participants in the CORTIS-01 analyses (i.e. inverse probability weighting). Enrolment in the CORTIS-HR study and Respiratory Pathobionts sub-study were independent of RISK11 status; analyses for these cohorts are not weighted.

For evaluation of diagnostic performance in the CTBC, CORTIS-HR, and Respiratory Pathobionts studies, area under the receiver operating characteristic (ROC) curve (AUC) was generated using the pROC package[34] in R. Signature performance (AUC) between HIV-infected and HIV-uninfected participants in the CTBC study was compared using methods described by DeLong et al.[35] Prognostic performance metrics through 15 months follow-up in CORTIS-HR were calculated by use of non-parametric methods for time-dependent ROC curve analysis from survival data using the R survAM.estimate function in the survAccuracyMeasures package[36] in R. Binary weighted ROC analysis was performed in the CORTIS-01 cohort to evaluate both diagnostic and prognostic performance. In a post-hoc analysis, prognostic performance in CORTIS-01 was qualitatively compared between 6, 12, and 15 months of follow-up for the primary endpoint by use of identical methods.

There were no pre-defined score thresholds for the parsimonious signatures; optimal signature score thresholds were calculated using the minimum Euclidean distance to 100% sensitivity and specificity, i.e. the point closest to the top-left of the ROC curve, calculated for each cut-point ($c$) as follows: $\sqrt{[(1 - \text{Sensitivity}_{(c)})^2 + (1 - \text{Specificity}_{(c)})^2]}$[37]. Signature diagnostic and prognostic accuracy was also benchmarked against the minimal and optimal WHO TPP for a community-based TB triage or referral test[20] and the minimal and optimal WHO TPP for an incipient TB test, to predict progression to TB disease[21], respectively.

Sensitivity, specificity, positive predictive value (PPV), and negative predictive value (NPV) at each threshold were calculated using binary endpoint indicators and standard formulae.

Estimates of PPV and NPV were computed based on the observed incidence of TB in the enroled study population. The 95% CIs on diagnostic and prognostic performance estimates were calculated with a non-parametric percentile bootstrap with 10,000 resamples[38]. Bootstrap sampling was stratified by RISK11 status in the CORTIS-01 cohort, but not stratified in the CTBC, CORTIS-HR, or Respiratory Pathobionts cohorts.

Signature score distribution was described using median and interquartile range. Differences between groups were calculated using the Mann–Whitney $U$ test and corrected for multiple comparisons by use of the Benjamini–Hochberg Procedure[39]. Spearman's rank coefficient was used to report correlation between signature scores; only samples with all assays passing were include in the correlation matrices; and Roe1 and Roe3 signatures were multiplied by −1 to obtain a positive correlation for all signatures. An alpha of <0.05 was considered significant in all analyses.

**Blinding**. Transcriptomic signature scores were measured by laboratory personnel who were blinded to participant TB status. Participants and study staff responsible for TB screening were blinded to transcriptomic signature scores. The statistical analysis team had access to clinical and demographic data (including TB status), but were blinded to signature scores, to allow data cleaning and preparation of analysis scripts prior to database lock. Signature scores and TB microbiology results were maintained in different files, which were only integrated after the study database had been cleaned and locked, and group allocations unblinded in CORTIS-01.

**Role of the funding sources**. The funders of this sub-study had no role in protocol development, study design, data collection, data analysis, data interpretation, or writing of the report.

## Results

### Transcriptomic signatures discriminate symptomatic pulmonary TB patients from healthy, asymptomatic controls.
We translated measurement of eight parsimonious TB transcriptomic signatures—Francisco2[25], Maertzdorf4[26] (DIAG4), Penn-Nicholson6 (RISK6)[23], Suliman4 (RISK4)[27], Roe1[28] (BATF2), Roe3[29], Sweeney3[30], and Thompson5[31] (RESPONSE5)—to a microfluidic RT-qPCR platform using TaqMan primer-probe assays for quantification of transcripts (Tables S1-6 in Supplementary Data 1). We then validated the performance of the signatures in the CTBC study, a case-control cohort of HIV-uninfected and HIV-infected individuals with or without pulmonary TB. Participants included 114 HIV-uninfected (53 TB cases and 61 asymptomatic controls) and 86 HIV-infected (45 TB cases and 41 asymptomatic controls) adults (Supplementary Fig. S1a). The performance of the Penn-Nicholson6 (RISK6) and RISK11 signatures in this cohort was previously reported[23], but is included here for comparison. We found that performance of all signatures to discriminate between HIV-uninfected South African adults with symptomatic pulmonary TB from healthy, asymptomatic controls was excellent, with AUC ranging from 0.89 (Thompson5, 95% confidence interval [CI] 0.82–0.95) to 0.97 (RISK11 and Suliman4, both with 95% CI 0.93–1) (Supplementary Fig. S2). All signatures met the minimal WHO triage test TPP benchmark criteria (sensitivity 90% and specificity 70%) in this group (Table S7A in Supplementary Data 1). Discriminative performance was generally lower in PLHIV, with AUCs ranging from 0.78 (Francisco2, 95% CI 0.68–0.88) to 0.96 (Suliman4, 95% CI 0.90–1), and was significantly worse (DeLong test $p < 0.05$) for RISK11 ($p = 0.027$), Francisco2 ($p = 0.0095$), and Sweeney3 ($p = 0.021$) (Supplementary Fig. S2). Only Penn-Nicholson6 and Suliman4 met the minimal WHO triage test TPP benchmark criteria

in PLHIV (Table S7B in Supplementary Data 1). Having validated the signatures for microfluidic RT-qPCR in a case-control context, we next wished to prospectively compare signature performance in a real-world setting.

### Recruitment of prospective community screening cohorts.
In the CORTIS-01 trial[11], 20,207 adults were screened at five geographically distinct community sites throughout South Africa between Sept 20, 2016, and Oct 19, 2018; 16,248 HIV-uninfected individuals met inclusion for the CORTIS-01 study, of whom 2923 participants were randomized and all were included in this parsimonious signature sub-study (Supplementary Fig. S1b and Table S8 in Supplementary Data 1). Between March 22, 2017, and May 15, 2018, a further 963 adults were screened and 861 PLHIV met inclusion for the CORTIS-HR study[12] and all were included in this sub-study (Supplementary Fig. S1d and Table S9 in Supplementary Data 1). There were 61 HIV-uninfected (weighted prevalence 1.1%, 95% CI 0.8–1.6) and 10 HIV-infected (prevalence 1.2%, 95% CI 0.6–2.1) baseline primary endpoint TB cases; over 80% of prevalent TB cases in the HIV-uninfected cohort, and 70% in the HIV-infected cohort, were asymptomatic. Thereafter, 362 HIV-uninfected participants (350 in the 3HP arm, 10 lost to follow-up, and 2 withdrawals) and 5 PLHIV (4 lost to follow-up and 1 withdrawal) were excluded from analysis of prognostic performance for incident TB, such that 2500 HIV-uninfected and 846 PLHIV were included. Through 15 months follow-up, 24 HIV-uninfected participants (weighted incidence 1.1 per 100 person-years, 95% CI 0.6–1.5) and 9 PLHIV (incidence 1.0 per 100 person-years, 95% CI 0.3–1.6) progressed to incident TB. Participants who discontinued follow-up prior to 15 months were included in the prognostic assessment, but censored at their final study visit.

### Measurement of parsimonious transcriptomic signatures in the prospective cohorts.
The RISK11 signature assay panel (Table S2 in Supplementary Data 1) was measured in all randomised HIV-uninfected CORTIS-01 trial participants ($N = 2923$) and in all enroled HIV-infected CORTIS-HR participants who had baseline RNA samples available 857/861 (99.5%). RISK11 performance in these cohorts was previously reported[11,12], but is included here for comparison without a pre-specified score threshold. The parsimonious transcriptomic signature assay panel (Table S3 in Supplementary Data 1) was measured in 2904/2923 (99.3%) HIV-uninfected and 858/861 (99.7%) HIV-infected participants who had additional baseline RNA aliquots available. The parsimonious signature failure rate ranged from 1.4% (RISK6) to 25.3% (Suliman4) in the HIV-uninfected cohort and from 5.2% (RISK6) to 32.1% (Suliman4) in PLHIV (Tables S10-11 in Supplementary Data 1; details in Supplementary Notes 2). Failed signature results for individual samples due to failed RT-qPCR reactions were not repeated, and were excluded from analysis.

Scores for all of the signatures were significantly correlated (Spearman-rank $p < 0.05$; Fig. 1); Parsimonious signature Spearman-rank correlation with RISK11 ranged from 0.41–0.90 in HIV-uninfected (Fig. 1a) and 0.25–0.88 in HIV-infected (Fig. 1b) cohorts. Signatures with overlapping genes, such as (1) Roe1, Roe3, and RISK11, and (2) Francisco2 and Sweeney3, were highly correlated (rho 0.84–0.96). The Thompson5 (RESPONSE5) treatment response monitoring signature had the poorest correlation with other signatures in HIV-uninfected (0.22–0.56) and HIV-infected (0.16–0.46) cohorts.

Median signature scores for RISK11, Francisco2, Maertzdorf4, RISK6, Suliman4, Sweeney3, and Thompson5 at baseline were highest amongst HIV-uninfected participants with symptomatic (clinical) TB, and successively lower in participants with

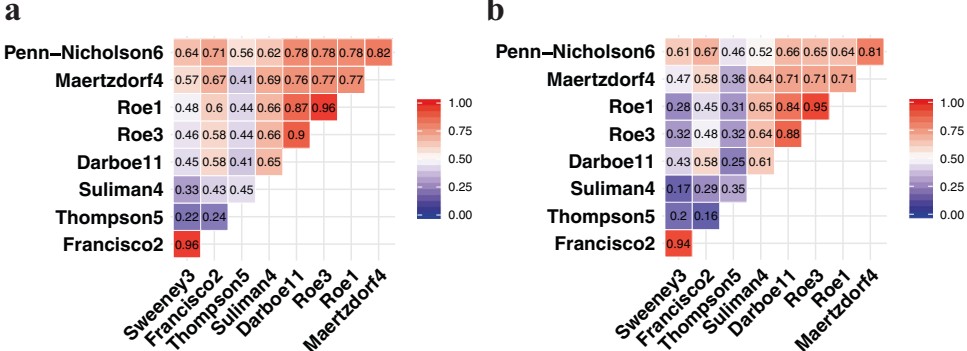

**Fig. 1 Correlation between parsimonious signature scores.** Signature score correlation matrix with the Spearman-rank-order correlation coefficients in (**a**) the CORTIS-01 study of people without HIV ($n = 2045^{\#}$) and (**b**) the CORTIS-HR study of people living with HIV ($n = 546^{\#}$). Roe1 and Roe3 signatures were multiplied by −1 to obtain a positive correlation for all signatures. #Only participants with signature scores available for all signatures were included in the Spearman correlation analysis.

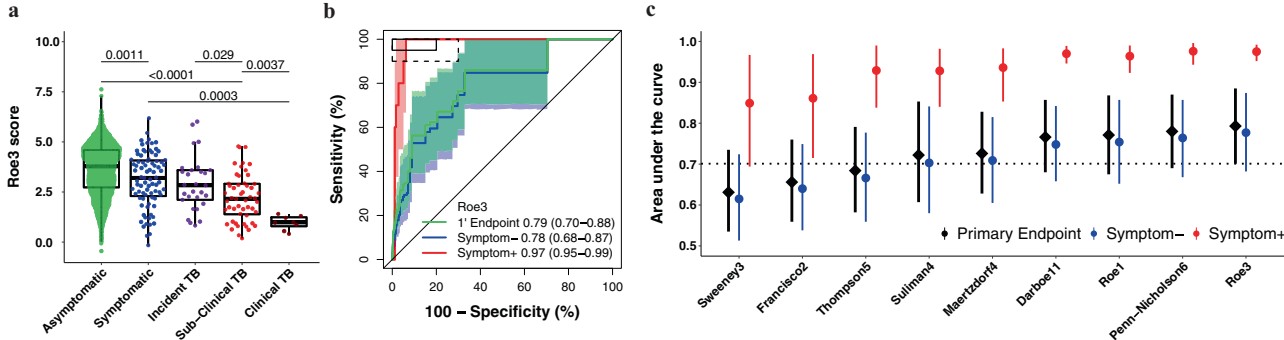

**Fig. 2 Parsimonious signature diagnostic performance for prevalent TB and signature score distributions in people without HIV.** Representative (**a**) box-and-whisker plot and (**b**) receiver operating characteristic (ROC) curve of the parsimonious signature with the best diagnostic performance (Roe3 signature) in the CORTIS-01 study of people without HIV. Box-and-whisker plots and ROC curves for the other signatures are in the Supplementary Information (Fig. S3). The box-and-whisker plot depicts Roe3 signature score (measured at enrolment) distribution by symptom status (each dot represents a participant) in asymptomatic ($n = 2570$) and symptomatic ($n = 83$) participants with no TB, participants who progressed to incident TB ($n = 29$), and participants with prevalent subclinical (asymptomatic; $n = 52$) and clinical (symptomatic; $n = 8$) TB. Prevalent and incident TB comprised all primary endpoint cases. Symptoms were recorded at the time of enrolment for participants without TB and those with prevalent TB. $p$-values for comparison of median signature scores between groups in the box-and-whisker plot were calculated with the Mann–Whitney $U$ test and corrected for multiple comparisons by use of the Benjamini–Hochberg Procedure[39]. Boxes depict the IQR, the midline represents the median, and the whiskers indicate the IQR ± (1.5 × IQR). The ROC curve depicts diagnostic performance (area under the curve, AUC, with 95% CI) of the Roe3 signature for the primary endpoint (1' Endpoint), i.e. TB diagnosed on two or more liquid culture-positive or Xpert MTB/RIF-positive sputum samples. The ROC curve shows participants with symptomatic clinical prevalent TB versus symptomatic controls (Symptom+), and participants with asymptomatic, subclinical prevalent TB versus asymptomatic controls (Symptom–). The shaded areas represent 95% CIs. The solid box depicts the optimal criteria (95% sensitivity and 80% specificity) and the dashed box depicts the minimal criteria (90% sensitivity and 70% specificity) set out in the WHO Target Product Profile for a triage test[20]. **c** Summary of signature diagnostic performance in the order of primary endpoint AUC estimates. The diagnostic AUC estimates in symptomatic and asymptomatic participant sub-groups are also shown. The midline indicates the AUC estimate, the error bars indicate the 95% CIs, and the black dotted line indicates the lower bound of the 95% CI for the best performing signature for the primary endpoint.

asymptomatic (subclinical) and incident TB (participants who progressed to TB disease), and symptomatic and asymptomatic participants without TB (Supplementary Fig. S3). The opposite was observed for the Roe1 and Roe3 signatures (Fig. 2 and Supplementary Fig. S3); this is related to data processing rather than real differences in signature biology, and a similar trend would be obtained by additively inverting the scores. A similar progression was observed for most signatures amongst PLHIV, although differences in the TB spectrum were not as marked, especially for Francisco2 and Sweeney3 (Fig. 3 and Supplementary Fig. S4).

**Similar diagnostic performance in HIV-uninfected and HIV-infected individuals, and inferior performance for detecting subclinical TB.** Diagnostic performance for primary endpoint

prevalent TB was moderate for all signatures, ranging from an AUC of 0.63 to 0.79 in the HIV-uninfected cohort (Fig. 2 and Table S10 in Supplementary Data 1). Primary endpoint diagnostic performance estimates in PLHIV (AUC 0.65–0.88) were similar or exceeded performance in the HIV-uninfected cohort for most signatures (Fig. 3 and Table S11 in Supplementary Data 1), except the Thompson5 and Sweeney3 signatures, which appeared to perform poorly in PLHIV (lower 95% confidence bound crossing 0.50). When disaggregated by TB symptom status, diagnostic performance for detection of asymptomatic (subclinical) TB among all asymptomatic participants, (AUCs ranging from 0.61 to 0.78) was inferior to performance for detection of symptomatic (clinical) TB among all symptomatic participants (AUCs ranging from 0.85–0.98) for all signatures in HIV-uninfected participants (Fig. 2). AUC estimates for detecting symptomatic TB were also

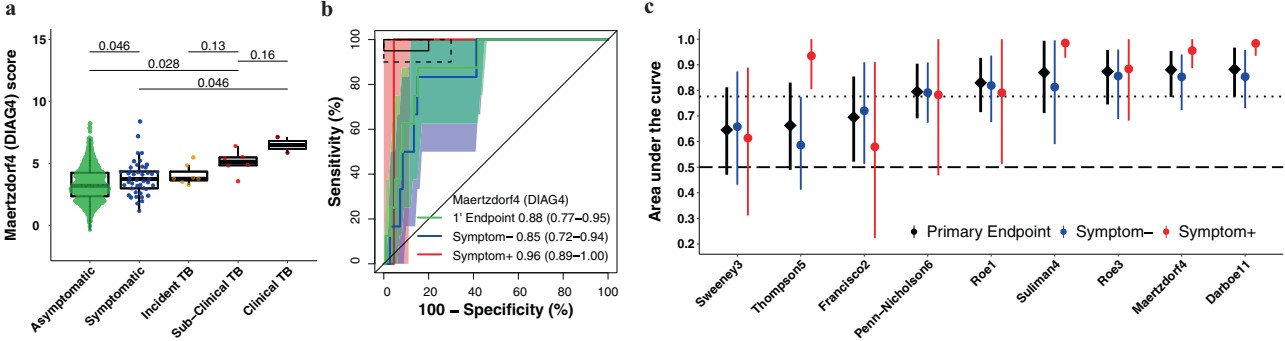

**Fig. 3 Parsimonious signature diagnostic performance for prevalent TB and signature score distributions in people living with HIV.** Representative (**a**) box-and-whisker plot and (**b**) receiver operating characteristic (ROC) curve of the parsimonious signature with the best diagnostic performance (Maertzdorf4/DIAG4 signature) in the CORTIS-HR study of people living with HIV. Box-and-whisker plots and ROC curves for the other signatures are in the Supplementary Information (Fig. S4). The box-and-whisker plot depicts Maertzdorf4 (DIAG4) signature score (measured at enrolment) distribution by symptom status (each dot represents a participant) in asymptomatic ($n = 743$) and symptomatic ($n = 45$) participants with no TB, participants who progressed to incident TB ($n = 7$), and participants with prevalent subclinical (asymptomatic; $n = 6$) and clinical (symptomatic; $n = 2$) TB. Prevalent and incident TB comprised all primary endpoint cases. Symptoms were recorded at the time of enrolment for participants without TB and those with prevalent TB. p-values for comparison of median signature scores between groups in the box-and-whisker plot were calculated with the Mann–Whitney $U$ test and corrected for multiple comparisons by use of the Benjamini–Hochberg Procedure[39]. Boxes depict the IQR, the midline represents the median, and the whiskers indicate the IQR ± (1.5 × IQR). The ROC curve depicts diagnostic performance (area under the curve, AUC, with 95% CI) of the Maertzdorf4 (DIAG4) parsimonious signature for TB diagnosed for the primary endpoint (1' Endpoint), i.e. TB diagnosed on two or more liquid culture-positive or Xpert MTB/RIF-positive sputum samples. The ROC curves show participants with symptomatic clinical prevalent TB versus symptomatic controls (Symptom+), and participants with asymptomatic subclinical prevalent TB versus asymptomatic controls (Symptom–). The shaded areas represent the 95% CIs. The solid box depicts the optimal criteria (95% sensitivity and 80% specificity) and the dashed box depicts the minimal criteria (90% sensitivity and 70% specificity) set out in the WHO Target Product Profile for a triage test[20]. **c** Summary of signature diagnostic performance in order of primary endpoint AUC estimates. The diagnostic AUC estimates in symptomatic and asymptomatic participant sub-groups are also shown. The midline indicates the AUC estimate, the error bars indicate the 95% CIs, and the black dotted line indicates the lower bound of the 95% CI for the best performing signature for the primary endpoint. The black dashed line indicates an AUC cut-off of 0.5.

higher for most signatures (RISK11, Maertzdorf4, Roe3, Suliman4, and Thompson5, with AUCs ranging from 0.58–0.98) amongst HIV-infected participants (Fig. 3) than for detecting asymptomatic, subclinical TB (AUCs ranging from 0.59–0.86). However, because only two symptomatic HIV-infected and eight symptomatic HIV-uninfected participants with prevalent TB had available RNA, these exploratory results should be interpreted cautiously. As the majority of healthy participants and TB cases were asymptomatic, the AUC estimates for the pooled primary endpoint are very similar to estimates for the asymptomatic subgroup. Diagnostic performance for the secondary endpoint was not qualitatively different from that of the primary endpoint for all signatures in both HIV-uninfected and HIV-infected cohorts (Supplementary Figs. S5-6).

For the primary diagnostic endpoint and asymptomatic sub-group, no signatures met the minimal WHO TPP benchmarks for a triage test in the HIV-uninfected cohort (Fig. 2, Supplementary Fig. S3 and Table S10 in Supplementary Data 1). However, five signatures (RISK11, RISK6, Roe1, Roe3, and Suliman4) met the optimal WHO TPP benchmark (sensitivity 95% and specificity 80%) amongst symptomatic participants, and the other four signatures (Francisco2, Maertzdorf4, Sweeney3, and Thompson5) either approached or met the minimal WHO TPP in this sub-group (Fig. 2 and Supplementary Fig. S3). Similarly amongst PLHIV, no signatures met the minimal WHO TPP benchmarks for a triage test (sensitivity 90% and specificity 70%) for the primary diagnostic endpoint and asymptomatic sub-group; however, four signatures (RISK11, Maertzdorf4, Suliman4, and Thompson5) met the optimal WHO triage test benchmarks in participants with TB symptoms (Fig. 3, Supplementary Fig. S4 and Table S11 in Supplementary Data 1). Due to the small numbers of primary endpoint prevalent TB cases with evaluable transcriptomic signature scores amongst PLHIV (two symptomatic and six asymptomatic), precision around estimates was

limited. For the primary diagnostic endpoint in the HIV-infected cohort, the upper limits of the 95% confidence intervals of performance metrics were within the bounds of the minimal WHO triage test criteria for most signatures, barring Francisco2, Sweeney3, and Thompson5 (Table S11 in Supplementary Data 1).

**Significant prognostic performance of signatures through 12 months in HIV-uninfected and 15 months in HIV-infected cohorts.** We next evaluated transcriptomic signature prognostic performance for incident TB in all participants without primary endpoint prevalent TB at baseline who attended one or more follow-up visit. All signatures measured at enrolment had excellent performance for detecting HIV-uninfected participants who progressed to primary endpoint incident TB disease through 6 months follow-up, with AUCs ranging from 0.80 to 0.95 (Fig. 4, Supplementary Fig. S7 and Table S12 in Supplementary Data 1). Prognostic performance declined through 15 months follow-up (AUCs 0.49–0.66) with only RISK6 showing significant discrimination at this timepoint (AUC 0.66, 95% CI 0.51–0.82). Most signatures demonstrated good discrimination (AUCs 0.63–0.80) up to 12 months post enrolment. In the HIV-infected CORTIS-HR cohort, there were insufficient primary endpoint incident TB cases to evaluate prognostic performance prior to 15 months follow-up. Six of nine transcriptomic signatures were able to significantly differentiate between incident TB cases and controls in the HIV-infected cohort through 15 months, with AUCs ranging from 0.54 to 0.81 (Fig. 5, Supplementary Fig. S8 and Tables S13 in Supplementary Data 1). Lower 95% confidence bounds for the Francisco2, Sweeney3, and Thompson5 signatures crossed 0.50 indicating that prognostic performance at this timepoint was not significant. Prognostic performance through 15 months follow-up for the secondary endpoint was not qualitatively different from the primary endpoint in either cohort (Supplementary Figs. S9–10).

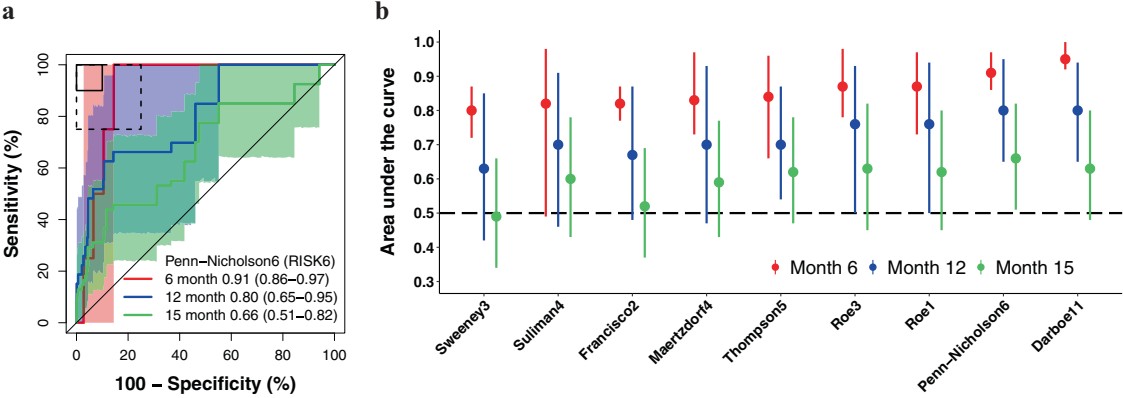

**Fig. 4 Parsimonious signature prognostic performance for incident TB in people without HIV. a** Representative receiver operating characteristic (ROC) curves depicting prognostic performance (area under the curve, AUC, with 95% CI) of the parsimonious signature with the best prognostic performance (Penn-Nicholson6/RISK6 signature) for incident TB diagnosed on two or more liquid culture-positive or Xpert MTB/RIF-positive sputum samples (primary endpoint) through 6, 12, and 15 months follow-up in the CORTIS-01 study of people without HIV. ROC curves for the other signatures are in the Supplementary Information (Fig. S7). The shaded areas represent the 95% CIs. The solid box depicts the optimal criteria (90% sensitivity and 90% specificity) and the dashed box depicts the minimal criteria (75% sensitivity and 75% specificity) set out in the WHO Target Product Profile for an incipient TB test[21]. **b** Summary of signature prognostic performance in the order of primary endpoint AUC estimates through 12 months follow-up. The prognostic AUC estimates through 6 and 15 months are also shown. The midline indicates the AUC estimate, the error bars indicate the 95% CIs, and the black dashed line indicates an AUC cut-off of 0.5.

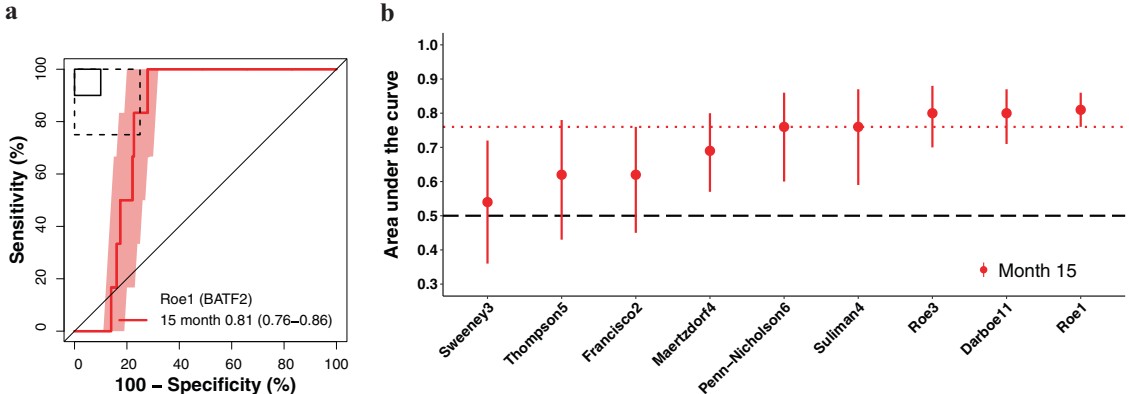

**Fig. 5 Parsimonious signature prognostic performance for incident TB in people living with HIV. a** Representative receiver operating characteristic (ROC) curve depicting prognostic performance (area under the curve, AUC, with 95% CI) of the parsimonious signature with the best prognostic performance (Roe1/BATF2 signature) for incident TB diagnosed on two or more liquid culture-positive or Xpert MTB/RIF-positive sputum samples (primary endpoint) through 15 months follow-up in the CORTIS-HR study of people living with HIV. ROC curves for the other signatures are in the Supplementary Information (Fig. S8). The shaded areas represent 95% CIs. The solid box depicts the optimal criteria (90% sensitivity and 90% specificity) and the dashed box depicts the minimal criteria (75% sensitivity and 75% specificity) set out in the WHO Target Product Profile for an incipient TB test[21]. **b** Summary of signature prognostic performance in the order of primary endpoint AUC estimates through 15 months follow-up. The midline indicates the AUC estimate, the error bars indicate the 95% CIs, the red dotted line indicates the lower bound of the 95% CI for the best performing signature for the primary endpoint, and the black dashed line indicates an AUC cut-off of 0.5.

For primary endpoint incident TB, none of the signatures met the minimal WHO TPP benchmark for a prognostic test (sensitivity 75% and specificity 75%) in the HIV-uninfected cohort through 15 months follow-up (Table S12 in Supplementary Data 1); the Roe3 signature came closest at its optimal cut-point (sensitivity 70.0%, 95% CI 46.0–92.9; specificity 72.9%, 95% CI 71.1–74.7). However, through 6 months follow-up, six of eight parsimonious signatures met the minimal benchmark. The remaining two signatures, Suliman4 and Sweeney3, approached it with upper limits of the 95% confidence intervals for prognostic estimates exceeding the benchmark bounds (Table S12 in Supplementary Data 1). Only RISK11 met the optimal WHO TPP benchmark for a prognostic test (sensitivity 90% and specificity 90%) through 6 months. Through 12 months follow-up, the upper limits of the 95% confidence intervals for all signatures exceeded the minimal WHO TPP benchmark, but only the

Roe3 signature (sensitivity 84.9%, 95% CI 55.0–100; specificity 75.4%, 95% CI 73.7–77.1) met the benchmark. Amongst people living with HIV, only the Roe1 signature (sensitivity 83.8%, 95% CI 28.1–98.6; specificity 78.0%, 95% CI 74.5–81.0) met the minimal WHO TPP benchmark for a prognostic test through 15 months follow-up (Table S13 in Supplementary Data 1); however the upper 95% confidence interval bounds for six of eight of the other signatures (except Maertzdorf4 and Sweeney3) also met or approximated this benchmark.

**Most TB signatures failed to differentiate participants with upper respiratory viral pathobionts from those with TB disease.** Amongst participants with HIV we found that those without TB who had a detectable HIV plasma viral load (≥100 copies/mL)

had significantly higher (Wilcoxon Rank-Sum test $p < 0.0001$; or conversely lower for Roe1 and Roe3) transcriptomic signature scores than those with supressed HIV plasma viral load (<100 copies/mL). The exception was Thompson5, the only signature that does not include ISGs[31], which was not affected by viral load ($p = 0.87$; Supplementary Fig. S11). Among HIV-uninfected participants without TB, scores for most signatures (except Francisco2 and Sweeney3) differed significantly (Wilcoxon Rank-Sum test $p < 0.05$) between those with and without TB symptoms (Fig. 2 and Supplementary Fig. S3); we hypothesised that these differences were due to the presence of other comorbid infections in the symptomatic participants.

We tested this hypothesis in a cohort 1000 HIV-uninfected CORTIS-01 participants who were consecutively enroled into the Respiratory Pathobionts sub-study (Supplementary Fig. S1c and Table S14 in Supplementary Data 1). Pathobionts encompass pathogenic and commensal microorganisms. All participants were investigated, irrespective of symptoms, for upper respiratory tract pathobionts in nasopharyngeal and oropharyngeal swabs using a multiplex bacteria and virus respiratory RT-qPCR panel. Viral and/or bacterial pathobionts were detected in 7.4% (74/1000) and 38.9% (389/1000) of participants respectively; with viral-bacterial co-detection in 3.3% (33/1000) participants.

TB transcriptomic signature scores were significantly higher (Wilcoxon Rank-Sum test $p < 0.05$; or conversely lower for Roe1 and Roe3) in participants with any viral pathobiont, or both viral and bacterial pathobionts, compared to those with only bacterial, or no pathobionts, with the exception of Thompson5 (Fig. 6 and Supplementary Fig. S12). Most of the signatures significantly discriminated between viral and bacterial pathobionts, and viral and no pathobionts, but were unable to differentiate bacterial from no pathobionts. We also measured the Herberg2 signature[32], designed to discriminate between viral and bacterial infection in febrile children, in the Respiratory Pathobionts cohort; this signature performed equivalently to the other signatures in differentiating between the groups. Most of the upper respiratory bacteria detected (*H influenza* 21.3%, *S aureus* 10.2%, *S pneumonia* 9.2%, *M catarrhalis* 5.8%) commonly colonise the respiratory tract of healthy individuals, hence we are unable to determine the effect of pathogenic bacteria on signature scores. Thompson5 was the only signature for which no differences in scores were observed between participants with viral, bacterial, or no pathobionts, and subsequently was not able to differentiate between these groups (Fig. 6).

28.6% (286/1000) of participants were co-enroled into the Respiratory Pathobionts sub-study and CORTIS-01 and consequently investigated for TB (Supplementary Fig. S1c). There were insufficient primary endpoint TB cases for meaningful analysis. However, 11 secondary endpoint (≥1 sputum sample positive for Mtb) prevalent TB cases (3.8%) were identified in CORTIS-01; 4/99 (4.0%) participants with bacteria detected in the upper respiratory tract and 7/150 (4.7%) with no pathobionts detected. No participants with viral upper respiratory pathobionts only ($n = 41$), or mixed viral and bacterial pathobionts ($n = 33$), had prevalent TB. The presence of bacterial upper respiratory pathobionts did not significantly affect TB diagnostic performance. Transcriptomic signatures differentiated between participants with prevalent TB and those with no pathobionts detected, with AUCs between 0.63 and 0.79, and between participants with prevalent TB and those with other bacterial pathobionts with AUCs ranging from 0.60 to 0.81 (Fig. 6 and Supplementary Fig. S13). However, with the exception of Thompson5 (AUC 0.73, 95% CI 0.51–0.93), none of the other signatures were able to differentiate between participants with prevalent TB and those with viral upper respiratory pathobionts.

## Discussion

We validated eight parsimonious TB transcriptomic signatures by microfluidic RT-qPCR and prospectively evaluated their diagnostic and prognostic accuracy in two large community screening cohorts of unselected HIV-uninfected and HIV-infected adults. We found that none of the signatures met the WHO triage test TPP benchmarks[20] for subclinical TB, although performance for several signatures approached these thresholds. Parsimonious signature diagnostic performance in adults with TB symptoms was superior to that in asymptomatic participants, with most meeting or approaching the TPP benchmarks. In the HIV-uninfected cohort prognostic accuracy for incident TB within 6 months of testing was excellent for virtually all parsimonious signatures, while performance for TB within 12 months was moderate. Only one signature, RISK6, had any statistically significant prognostic ability for incident TB within 15 months of testing.

This waning accuracy mirrors the finding of a short prognostic window after signature testing in the meta-analysis by Gupta et al.[14], and supports the hypothesis that the timeframe of TB progression from infection to clinical disease is highly heterogenous between individuals, ranging from months to years[40–42]. We postulate that the declining performance distal to signature measurement may be due to low inflammatory activity in early disease stages, described as incipient TB. Individuals with Mtb replication (incipient TB) may spontaneously control the infection and thus not progress to disease[41], resulting in reduced specificity. In this regard, we suggest that the 2-year predictive window for a prognostic test proposed in the WHO framework[21] be revisited. Transient viral infections may also increase false positivity[24]. Low sensitivity may reflect the high burden of population Mtb exposure and rate of new infections in this endemic setting.

Performance of parsimonious signatures in PLHIV appeared to match or surpass performance in participants without HIV, consistent with the findings of Turner and colleagues[15]. Most signatures were able to differentiate between those who would progress to TB disease (progressors) through 15 months follow-up and non-progressors in the HIV-infected cohort, and the Roe1 signature met the minimal WHO TPP benchmark for a prognostic test[21]. The greater prognostic window of the signatures measured at baseline in PLHIV may reflect prevalent TB cases missed at study enrolment (possibly due to paucibacillary sputum), but detected further along the disease spectrum during follow-up. At each study visit, participants underwent symptom screening with symptom-guided TB investigations which would have missed asymptomatic disease at earlier timepoints. However, all participants provided sputum samples for TB investigation at the end-of-study visits, which may also explain the bulk of disease diagnosed at the month 15 visit. Most of the incident TB cases in the HIV-infected cohort were detected after 12 months follow-up; this may be due to the appearance of symptoms late in the disease course in immunosuppressed individuals, and as a result, incident TB cases may have been missed by symptom screening at earlier visits. It is also conceivable that the better performance in PLHIV is driven by ART-naïve individuals who have increased risk of TB disease, but also have elevated signature scores (or lower for Roe1 and Roe3) due to their higher HIV plasma viral loads. However, 74% (142/193) of ART-naïve participants started ART during the study (Table S9 in Supplementary Data 1), which would lower their risk of progression, and increase false positivity. Unfortunately our study was underpowered to evaluate signature performance stratified by viral load or ART status, however we postulate that higher signatures scores in those with non-supressed HIV plasma viral loads (Supplementary Fig. S11) would result in lower specificity, but equivalent or higher sensitivity.

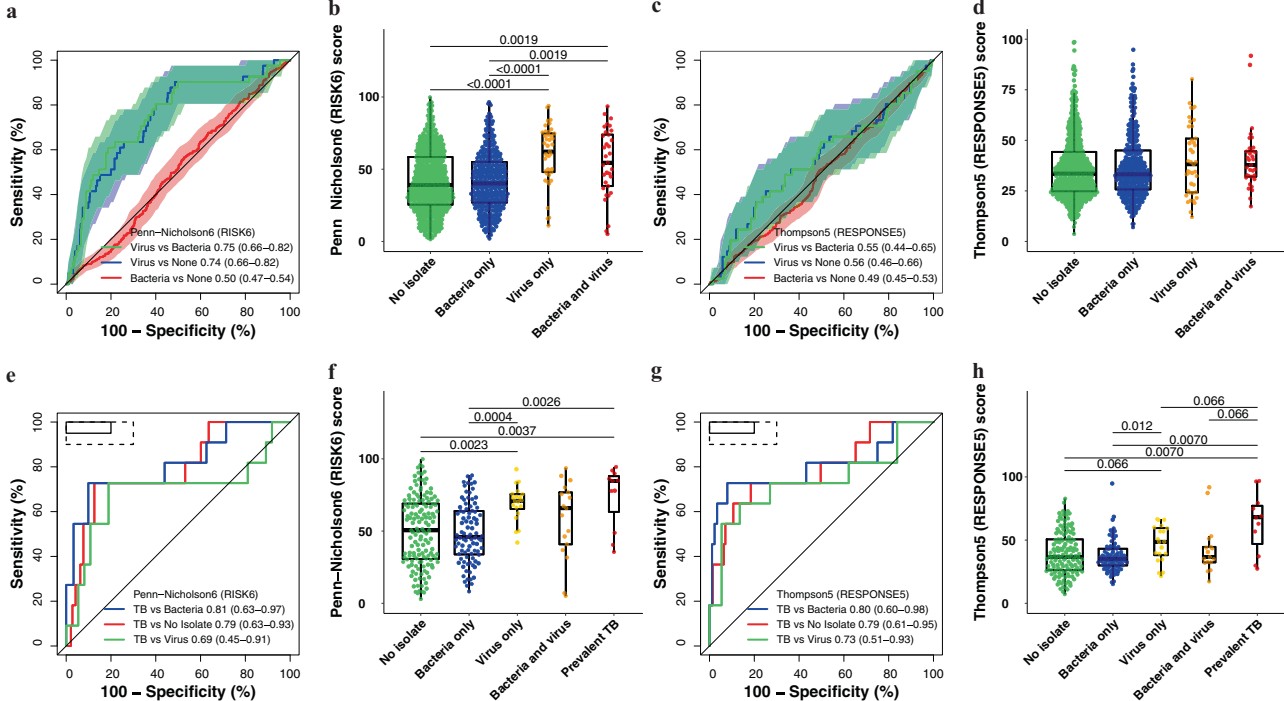

**Fig. 6 Parsimonious signature performance for differentiating participants with viral or bacterial upper respiratory tract pathobionts, those without any pathobionts, and participants with prevalent TB.** Representative receiver operating characteristic (ROC) curves and box-and-whisker plots for the (**a**, **b**) Penn-Nicholson6 (RISK6) and (**c**, **d**) Thompson5 (RESPONSE5) signatures in all participants without prevalent TB in the Respiratory Pathobionts substudy (11 participants with secondary endpoint prevalent TB excluded). ROC curves and box-and-whisker plots for the other signatures are in the Supplementary Information (Fig. S12). The box-and-whisker plots depict signature score distributions in participants with no upper respiratory pathobionts ($n = 563$), bacterial upper respiratory pathobionts only ($n = 352$), viral upper respiratory pathobionts only ($n = 41$), both viral and bacterial upper respiratory pathobionts ($n = 33$). The ROC curves depict performance (area under the curve, AUC, with 95% CI) of the parsimonious signatures in differentiating between participants with viral upper respiratory pathobionts and participants with bacterial upper respiratory pathobionts, between participants with viral pathobionts and those with no pathobionts, or between participants with bacterial pathobionts and those with no pathobionts. The shaded areas represent the 95% CIs. Representative ROC curves and box-and-whisker plots for the (**e**, **f**) Penn-Nicholson6 (RISK6) and (**g**, **h**) Thompson5 (RESPONSE5) signatures in all participants randomised in the CORTIS-01 study and co-enroled in the Respiratory Pathobionts sub-study. Participants who were not randomised in the CORTIS-01 study, and thus not investigated for TB, are not included. ROC curves and box-and-whisker plots for the other signatures are in the Supplementary Information (Fig. S13). The ROC curves depict performance of the parsimonious signatures in differentiating between participants with prevalent TB ($n = 11$) and participants with viral upper respiratory pathobionts ($n = 37$), participants with bacterial upper respiratory pathobionts only ($n = 95$), and participants with no pathobionts ($n = 143$). Participants with both viral and bacterial pathobionts ($n = 17$) were included in the group with viral pathobionts only ($n = 20$) as the presence of bacterial pathobionts did not appear to affect signature scores. The solid box depicts the optimal criteria (95% sensitivity and 80% specificity) and the dashed box depicts the minimal criteria (90% sensitivity and 70% specificity) set out in the WHO Target Product Profile for a triage test[20]. The box-and-whisker plots depict signature score distribution in participants with no upper respiratory pathobionts, bacterial upper respiratory pathobionts only, viral upper respiratory pathobionts only, both viral and bacterial upper respiratory pathobionts, and *Mycobacterium tuberculosis* detected on GeneXpert MTB/RIF or MGIT culture (microbiologically confirmed secondary endpoint prevalent TB; i.e. TB confirmed on at least one sputum sample). p-values for comparison of median signature scores between groups in box-and-whisker plots were calculated with the Mann–Whitney *U* test and corrected for multiple comparisons by use of the Benjamini–Hochberg Procedure[39]. Only p-values below 0.1 are shown. Each dot represents a participant. Boxes depict the IQR, the midline represents the median, and the whiskers indicate the IQR ± (1.5 × IQR).

We recently showed that elevated RISK11 signature scores in healthy individuals without TB return to normal levels within three months irrespective of the use of TPT in the majority of individuals[24]. The transient elevation of RISK11 may be due to self-clearance of Mtb infection in certain individuals, but is likely due to other exogenous factors that temporarily affect transcriptomic signature gene expression. Scores and performance of most parsimonious signatures, with the exception of the Thompson5 signature, were affected by the presence of upper respiratory viral, but not bacterial, pathobionts. Most TB transcriptomic signatures primarily detect elevated ISG expression, which are known to be upregulated as part of an inflammatory response typically induced in TB and viral infections, including HIV[10,12,24]. The Thompson5 signature, which does not contain

any ISGs, was unaffected by respiratory pathobionts or HIV. This supports the hypothesis that exogenous factors, and in particular viral infections, have a large impact on TB transcriptomic signature scores and performance due to induction of these genes; design of transcriptomic signatures which do not predominantly consist of ISGs[43,44], or which use a multinomial modelling[45] approach to exclude viral infection, may address this shortcoming.

Given that this study was not designed to compare diagnostic and prognostic accuracy between signatures, and the 95% confidence intervals of performance metrics for most signatures were overlapping, we are hesitant to champion any particular signature. Several parsimonious signatures performed at least as well as RISK11. Excellent NPV (>99%) for all transcriptomic

biomarkers in the CORTIS studies make these good rule-out tests, however low PPV (<6%), reflective of low TB prevalence and incidence in this community setting, make these inadequate rule-in tests. Performance of most signatures is promising for triage of symptomatic adults seeking care and PLHIV attending ART clinic, to rule-out active disease and guide further investigation (e.g. sputum Xpert Ultra or radiography), and for short-term (6–12 months) prognosis of TB disease risk for targeted preventive therapy in those without disease. Recent studies of 3HP have shown limited duration of efficacy (9 months) for biomarker-guided preventive therapy in HIV-uninfected adults in South Africa[11], and no additional benefit of universal annual retreatment of PLHIV on ART compared to a single course[46]. Our results further suggest that a prognostic window greater than 12 months after testing is unlikely to be of value in high transmission settings with associated risk of re-infection. This implies that annual re-testing would likely be necessary if such signatures are to be included in TB control strategies[47]. Parsimonious signature performance reported here warrants translation to point-of-care devices for clinical implementation studies.

No signatures met the WHO triage test TPP accuracy benchmark for diagnosing asymptomatic prevalent TB, which is a setback for community active case-finding efforts in endemic settings, such as South Africa, where there is a high burden of undiagnosed subclinical disease[2–5,11,12]. However, we would note that these benchmarks were not intended for TB screening in a healthy community setting, but rather "to test persons with any symptoms or risk factors suggestive of TB who are seeking care"[20], and thus the bar may have been too high for a strategy that seeks to find subclinical TB cases. Currently, chest radiography is the only sufficiently accurate screening tool in widespread use for detecting asymptomatic subclinical disease[2–5]. Portable molecular screening has also been proposed as a possible universal screening tool for active case-finding[7], but this remains unaffordable in most settings. Given the recently recognised burden of asymptomatic TB in endemic settings and lack of affordable and accessible active case-finding tools to detect early and subclinical disease in community settings, we would propose that a new consensus framework for the development of screening tests for asymptomatic TB is urgently needed. Such a framework should also provide clear definitions of contentious concepts of asymptomatic incipient and subclinical TB[48,49]. We have previously demonstrated a sequential inflammatory TB disease process[23,40,50]. The trend of increasing signature scores from individuals without TB disease, through early or incipient (individuals who later progress to incident disease), subclinical, and clinical TB disease observed in this study (Figs. 2–3) succinctly illustrates the stages of disease progression. Further trials are needed to determine whether earlier stages of TB progression require a full course of curative treatment, or an abbreviated regimen such as 3HP.

The correlation between most signatures (with the exception of Thompson5), and fact that no individual signature was significantly superior for diagnosis or prognosis, indicates they likely measure the same or similar biological pathways. Our study would suggest that performance of ISG dominant transcriptomic host-response signatures may be approaching an apex and that further signature discovery using similar conventional methodology and study designs is unlikely to deliver better diagnostic tools. What could be done differently? First, we would argue the necessity for recruiting larger heterogenous unselected prospective discovery cohorts, with TB cases representing a spectrum from subclinical through clinical disease at prevalence rates that reflect the real epidemic, rather than carefully-curated case-control studies with uniform and highly enriched TB cases and homogenous healthy controls or inappropriate other diseases. In this study, all signatures performed superbly in the CTBC case-control cohort, but performance dipped substantially in the real-world CORTIS studies. Second, we have shown that parsimonious signatures perform at least as well as non-reduced signatures such as RISK11; early translation of parsimonious signatures to near point-of-care technology, such as RT-qPCR, would allow more rapid advancement along the developmental pipeline. Third, few studies have incorporated clinical features (such as age, sex, body-mass index, C-reactive protein, haemoglobin, monocyte-lymphocyte ratio, and HIV status or viral load) into predictive host-response models, which may prove low-hanging fruit for improving signature performance[51,52]. Finally, selection of genes which remain stable on repeat measurements through longitudinal follow-up in healthy individuals[24,50], and in individuals with common comorbid diseases and infections, may enhance predictive utility. Modular (genes selected from distinct biological pathways) or multinomial (multiple outcomes or covariates) modelling approaches to TB signature discovery may help to mitigate the effect of exogenous factors on signature performance[43,45].

A major strength of this study was the prospective recruitment of large community cohorts of unselected adults from multiple sites reflecting distinct patterns of Mtb exposure and rates of TB disease in an endemic setting. Despite intensive community active case-finding with enrichment of the HIV-uninfected study population for individuals at greatest risk of TB disease, sputum sample collection from all enroled participants at enrolment and end-of-study visits, and symptom-triggered investigation during follow-up, there were relatively few prevalent and incident TB cases, reducing the precision of sensitivity and PPV estimates. We also did not investigate participants for extra-pulmonary TB, potentially missing cases amongst PLHIV. Another limitation was the high Suliman4 failure rate, resulting in less precise performance estimates for this signature. Since we put together our signature panel in 2017, numerous new TB transcriptomic signatures have been discovered, including several in cohorts with symptomatic controls with other diseases. It is possible that other parsimonious signatures, or derivatives of larger signatures, may measure other aspects of the immune response to TB which are less affected by viral infection (as demonstrated by Thompson5), and may have greater diagnostic or prognostic utility, than the predominantly ISG signatures included in this study. We also acknowledge that primer sequences selected for RT-qPCR may not be equivalent to those used for microarray or in the Cepheid Xpert MTB Host Response prototype cartridge[17–19]. The large proportion of asymptomatic prevalent TB cases is a feature unique to this cohort, and may prove valuable for future biomarker discovery. Recruitment was restricted to South Africa, limiting generalisability of the results to other contexts; there is a need for similar large prospective diagnostic validation cohorts in other TB endemic regions.

Several parsimonious TB transcriptomic signatures hold promise for triage of symptomatic adults seeking care, screening of ART clinic attendees and other high-risk groups for further investigation, and prediction of short-term risk of TB for initiation of targeted preventive therapy. Point-of-care RT-qPCR testing platforms currently under development and in field implementation studies[17–19], would bring these signatures from the laboratory bench to the clinic bedside.

**Reporting summary.** Further information on research design is available in the Nature Research Reporting Summary linked to this article.

## Data availability

Deidentified signature scores, clinical metadata, and TB endpoint data are provided in **S15-18** in Supplementary Data 1. The public PCR probe dataset and metadata have been deposited in Zivahub (https://doi.org/10.25375/uct.14999895), an open access data repository hosted by the University of Cape Town's institutional data repository powered by Figshare for Institutions[53].

## Code availability

The locked-down R code used to analyse the Fluidigm 192.24 gene expression chips, with quality control filters that assessed the integrity and reproducibility of each chip, is available at bitbucket.org/satvi/sixs. See further details in the Methods.

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

## Acknowledgements

We acknowledge the substantial contributions of The Cross-sectional TB Cohort (CTBC) Study Team and The CORTIS Study Team (lists of consortia members and their affiliations appear in the Supplementary Information). The CTBC study was funded by the Strategic Health Innovation Partnerships (SHIP) Unit of the South African Medical Research Council (SAMRC) with funds received from the South African Department of Science and Technology. The CORTIS studies were funded by the Bill and Melinda Gates Foundation (BMGF; OPP1116632, OPP1137034, and OPP1151915) and also the SHIP Unit of the SAMRC with funds received from the South African Department of Science and Technology. Respiratory infection assays were funded by the NIH (AI123780). SCM is a recipient of PhD funding from the South African Medical Association (SAMA), the Fogarty International Center of the National Institutes of Health (NIH) under Award Number D43 TW010559, the Harry Crossley Clinical Research Fellowship, and the South African Medical Research Council (SAMRC) through its Division of Research Capacity Development under the SAMRC Clinician Researcher Programme. The content is solely the responsibility of the authors and does not necessarily represent the official views of SAMA, the NIH, the Harry Crossley Foundation, or the SAMRC. The funders had no role in study design, data collection and analysis, decision to publish, or preparation of the manuscript.

## Author contributions

S.C.M., A.P.-N., A.F.-G., M.H., and T.J.S. conceived the study. A.P.-N., S.K.M., M.H., and T.J.S. implemented clinical studies, raised funds, and/or provided the resources. M.T., G.W., K.N., and G.C. were responsible for all site-level activities, including recruitment, clinical management, and data collection. S.K.M., M.M., H.M., M.F., K.H., and M.E., and provided operational, technical, or laboratory support and project management. S.C.M., M.E., and O.N. processed samples and performed the experiments. S.C.M., S.K.M., A.F.-G., and H.M. verified the underlying data. S.C.M. and A.F.-G. analysed data. S.C.M., S.K.M., T.J.S., and M.H. interpreted results and wrote the first draft of the manuscript. Members of the CTBC Study Team and the CORTIS Study Team performed clinical studies and processed samples. All authors had full access to the data, confirm the integrity of the data and its presentation, agree with its interpretation as discussed in the manuscript, and reviewed, revised, and approved the manuscript before submission. The corresponding author had final responsibility for the decision to submit for publication.

## Competing interests

A.P.-N., G.W., G.C., T.J.S., and M.H. report grants from the Bill & Melinda Gates Foundation, during the conduct of the study; A.P.-N., K.N., and G.W. report grants from the South African Medical Research Council, during the conduct of the study; K.N. reports a grant from the US Centers for Disease Control and Prevention, G.W. and T.J.S. report grants from the South African National Research Foundation, during the conduct of the study. In addition, A.P.-N. and T.J.S. have a patent of the RISK11 (Darboe11) and RISK6 (Penn-Nicholson6) signatures issued; G.W. and T.J.S. have a patent of the RISK4 (Suliman4) signature pending. G.W. has a patent "TB diagnostic markers" (PCT/IB2013/054377) issued and a patent "Method for diagnosing TB" (PCT/IB2017/052142) pending. All other authors had nothing to disclose.

## Additional information

[1]South African Tuberculosis Vaccine Initiative, Institute of Infectious Disease and Molecular Medicine, and Division of Immunology, Department of Pathology, University of Cape Town, 7925 Cape Town, South Africa. [2]Vaccine and Infectious Disease Division, Fred Hutchinson Cancer Research Center, Seattle, WA 98109, USA. [3]DST/NRF Centre of Excellence for Biomedical TB Research; South African Medical Research Council Centre for TB Research; Division of Molecular Biology and Human Genetics, Department of Biomedical Sciences, Faculty of Medicine and Health Sciences, Stellenbosch University, 7505 Cape Town, South Africa. [4]Centre for the AIDS Programme of Research in South Africa (CAPRISA), 4001 Durban, South Africa. [5]MRC-CAPRISA HIV-TB Pathogenesis and Treatment Research Unit, Doris Duke Medical Research Institute, University of KwaZulu-Natal, 4001 Durban, South Africa. [6]The Aurum Institute, 2194 Johannesburg, South Africa. [7]School of Public Health, University of Witwatersrand, 2193 Johannesburg, South Africa. [8]Department of Medicine, Vanderbilt University, Nashville, TN 37232, USA. ✉email: Thomas.Scriba@uct.ac.za

