## [Peer Review File · Communications Medicine]

This manuscript has been previously reviewed at another Nature Portfolio journal. This document only contains reviewer comments and rebuttal letters for versions considered at Communications Medicine.

Reviewers' comments:

Reviewer #1 (Remarks to the Author):

I have no issues with the revised manuscript.

Reviewer #2 (Remarks to the Author):

The reviewers have provided a very comprehensive rebuttal and have addressed most of my concerns with this manuscript. Thank you for the clarification on data availability. I also think the rationale for focusing on 6-gene signatures developed before 2017 is justified, but is a major limitation of the study. I think the authors have clarified the 2017 selection in the methods, and with the changes in the discussion (ISG signatures reaching apex, mentioning other signatures can be developed) further alleviate my concerns. I would also like to see the Intro or beginning of the results to mention the 2017 selection as well. Other than that change, I have no remaining concerns.

Reviewer #3 (Remarks to the Author):

The authors have responded to the questions raised in the last review. A few issues remain that require some revision

1. The amount of new information is limited. The figures are large and complicated but of less value to the reader than the conclusions - rather than show all of the ROC curves a concise presentation of summary data would be preferable
2. There is inconsistency as to whether TPP exists for asymptomatic TB - it is worth noting that the triage strategy coupled with active case finding will allow identification of infectious TB cases that otherwise would not be diagnosed or treated at an early stage. The cost-effectiveness and TPP may differ considerably compared to symptomatic TB
3. The PPV noted may be cost-prohibitive for application in the community setting - 94+% of those that are positive require a confirmatory diagnostic, etc. The settings in which the test is likely to be a suitable triage test should be discussed
4. The "positives" found in viral infection including HIV with high viral loads are considered to be "false positives" - it is possible however that these conditions are associated with increased MTB replication - again this should be considered
5. The suggestion that the window for TB be shortened to improve specificity is controversial - the ideal prognostic test would identify late as well as early cases

Reviewer #1:

I have no issues with the revised manuscript.

Reviewer #2:

The reviewers have provided a very comprehensive rebuttal and have addressed most of my concerns with this manuscript. Thank you for the clarification on data availability.

C1: I also think the rationale for focusing on 6-gene signatures developed before 2017 is justified, but is a major limitation of the study. I think the authors have clarified the 2017 selection in the methods, and with the changes in the discussion (ISG signatures reaching apex, mentioning other signatures can be developed) further alleviate my concerns. I would also like to see the Intro or beginning of the results to mention the 2017 selection as well. Other than that change, I have no remaining concerns.

R1: The following sentence was moved from Methods to Results (lines 115-118): "The signatures were pragmatically selected in 2017 for inclusion in this head-to-head validation alongside RISK11 based on availability of validated performance data, number of transcripts (six or fewer), and availability of target sequences for custom primer-probe design or predesigned TaqMan assay."

Reviewer #3:

The authors have responded to the questions raised in the last review. A few issues remain that require some revision

C2: The amount of new information is limited. The figures are large and complicated but of less value to the reader than the conclusions - rather than show all of the ROC curves a concise presentation of summary data would be preferable.

Editor: With regards the request from Reviewer #3 that you remove the ROC curves, we ask instead that you include these as SI and add the requested data summary to the main manuscript.

R2: We have moved most of the ROC curves and violin plots to the Supplement as requested. We have kept the summary figure, and a representative ROC curve and violin plot of the best performing concise signature (excluding RISK11, since that is already reported in Scriba et al., *Lancet Infect Dis* 2021 or Mendelsohn et al., *Lancet Glob Health* 2021), for each figure in the main manuscript. We also provide this rationale in each figure legend.

C3: There is inconsistency as to whether TPP exists for asymptomatic TB - it is worth noting that the triage strategy coupled with active case finding will allow identification of infectious TB cases that otherwise would not be diagnosed or treated at an early stage. The cost-effectiveness and TPP may differ considerably compared to symptomatic TB.

R3: We agree with the reviewer. The importance of such a triage strategy for earlier diagnosis is highlighted in the introduction (lines 60-64). The lack of a specific TPP for asymptomatic/subclinical TB is discussed in lines 390-402. We feel that cost-effectiveness of such a strategy is beyond the scope of this paper.

C4: The PPV noted may be cost-prohibitive for application in the community setting - 94+% of those that are positive require a confirmatory diagnostic, etc. The settings in which the test is likely to be a suitable triage test should be discussed.

R4: The interpretation of the low PPV and discussion of settings in which the test is likely to be a suitable triage test are in lines 375-381 and lines 457-459. As mentioned in the point above, cost-effectiveness of such a strategy is beyond the scope of this paper.

C5: The "positives" found in viral infection including HIV with high viral loads are considered

to be "false positives" - it is possible however that these conditions are associated with increased MTB replication - again this should be considered

R5: We have amended lines 327-329 as follows: "Individuals with **Mtb replication** (incipient TB) may spontaneously control the infection and thus not progress to disease³², resulting in reduced specificity."

C6: The suggestion that the window for TB be shortened to improve specificity is controversial - the ideal prognostic test would identify late as well as early cases

R6: We agree with the reviewer that an ideal prognostic test would identify late as well as early cases, however our data suggests that this may not be feasible. We discuss this in lines 318-332.

REVIEWERS' COMMENTS:

Reviewer #3 (Remarks to the Author):

No further comments. MS much improved in the review